# Post-hospital physical rehabilitation for physical function recovery among community-dwelling survivors of critical illness: A systematic review protocol

Christopher Farley[1], Anastasia N.L. Newman[1], Christine Caron[2], Kimia Honarmand[3], Stuart M. Phillips[4], Jenna Smith-Turchyn[1], Dina Brooks[1,5]*

1 Faculty of Health Science, School of Rehabilitation Science, McMaster University, Hamilton, Ontario, Canada, 2 Sepsis Canada, Hamilton, Ontario, Canada, 3 Division of Critical Care, Department of Medicine, McMaster University, Hamilton, Ontario, Canada, 4 Department of Kinesiology, McMaster University, Hamilton, Ontario, Canada, 5 Department of Respiratory Medicine, West Park Healthcare Centre, Toronto, Ontario, Canada

* brookd8@mcmaster.ca

## Abstract

### Introduction

Many survivors of critical illness experience lasting physical disability. Post-hospital rehabilitation has the potential to reduce this physical disability; however, primary studies have reported inconsistent results. We aimed to answer the following question: Among community-dwelling adults who survived critical illness, does participation in post-hospital physical rehabilitation, compared with no rehabilitation or alternative non-physical interventions, improve physical functioning 12 months after discharge from acute care?

### Materials and methods

This is a systematic review protocol that was registered with PROSPERO (CRD 420251174065). It will be conducted with Cochrane methods and reported according to the 2020 Preferred Reporting Items for Systematic Review and Meta-analysis (PRISMA) statement. To optimize the impact of this review, our study team includes a patient partner to guide our methods and interpretation. To be eligible, peer-reviewed randomized controlled trials must have enrolled community-dwelling adults (≥ 18 years) previously admitted to the ICU (≥24 hours) to a post-hospital physical rehabilitation program compared to any control. Outcomes of interest include physical function, return to work, and health care utilization. We will search five databases from their inception. Study screening, selection and extraction will be conducted independently and in duplicate using Covidence. Disagreements will be resolved through discussion or with a third reviewer. We will assess risk of bias using version 2 of the

**Data availability statement:** No datasets were generated or analyzed during the current study. All extracted data from this study will be made available on Open Science Framework upon study completion.

**Funding:** CF is partially funded by the Canadian Lung Association Allied Health Research Fellowship to complete this work. The Canadian Lung Association had no part in the study design of this protocol and will have no role in data collection and analysis, decision to publish or preparation of the resulting systematic review manuscript. No additional external funding was received for this study.

**Competing interests:** No authors have competing interests.

Cochrane risk-of-bias tool for randomized trials (RoB 2). Where appropriate, we will conduct meta-analyses using random-effects modeling. Certainty of evidence will be assessed using the Grading of Recommendations Assessment, Development and Evaluation (GRADE).

## Conclusion

A recent international multiprofessional expert panel highlighted the importance of understanding how follow-up care models can optimize long-term recovery after critical illness. Our review will provide a comprehensive synthesis of the impact of post-hospital rehabilitation on long-term recovery of survivors of critical illness.

---

## Introduction

The intensive care unit (ICU) provides care for people who have or are at risk of life-threatening illness, particularly of the respiratory, cardiac and renal systems [1]. With advances in intensive care medicine knowledge and technology, survival rates from critical illness are improving. Despite higher illness severity among patients, a retrospective time series analysis found a 35% decrease in ICU mortality from 1988 to 2012 in United States' hospitals [2]. With improved survival, intensive care approaches no longer focus solely on survival but now integrate approaches to reduce the negative sequelae associated with critical illness [3].

Post-intensive care syndrome is a new onset or worsening of impairments in physical, cognitive and/or mental health that arises from an ICU admission and persists beyond hospital discharge [4,5]. These impairments are closely linked to the profound muscle loss experienced by adults with critical illness, who lose 2.1% (95% confidence interval [CI] 1.0 to 3.2) rectus femoris and 2.2% (95% CI 1.8 to 2.6) biceps brachii cross-sectional area each day [6]. Furthermore, for each day of immobilization within the ICU, survivors experienced a 3% (95% CI 0–7) reduction in muscle strength at hospital discharge [7]; this loss was compounded such that patients experienced an 11% (95% CI 4–19) strength reduction at 2-year follow-up for each day of ICU bedrest [7]. Overall, these physical impairments contribute to the 40% of people with critical illness who develop ICU-acquired weakness [8].

Seminal research followed survivors of critical illness after hospital discharge and found that 1 year after hospital discharge, they experienced physical disability with 6-minute walk test (6MWT) scores that were 66% of age- and sex-matched norms [9]. Even 5 years after hospital discharge, these survivors continued to have physical disability with 6MWT scores that were 76% of age- and sex-matched norms [10]. Further research has found lasting impacts on participation among survivors of critical illness. A cohort of 5,762 survivors of critical illness found that 2 years after hospital discharge, 30% were unable to return to work [11]. Post-hospital rehabilitation has the potential to reduce the physical disability and resultant participation limitations associated with critical illness.

Recent syntheses have indicated the potential benefit of post-hospital rehabilitation [12,13]; compared to controls, Cazeta et al. [12] reported significant improvement in SF-36 Physical Function scores among four trials (N = 341, mean difference = 4.04 [1.19 to 6.90]). However, these recent reviews only included patients who had required invasive mechanical ventilation [12,13], which only represents about one-third of those admitted to the ICU [14]. To capture the overall effect of post-hospital rehabilitation on those after critical illness, broad population eligibility criteria are necessary. As such, we set out to answer the following question: Among community-dwelling adults who survived critical illness, does participation in post-hospital physical rehabilitation, compared with no rehabilitation or alternative non-physical interventions, improve physical functioning 12 months after discharge from acute care?

## Materials and methods

We registered our systematic review with the International Prospective Register of Systematic Reviews (PROSPERO) on 29 October 2025 (CRD420251174065). We have reported our protocol according to the Preferred Reporting Items for Systematic Review and Meta-analysis (PRISMA) Protocols (PRISMA-P) 2015 statement (S1 Table in S1 File) [15]. Our systematic review will be conducted using methods consistent with the Cochrane Handbook for Systematic Reviews of Interventions [16] and reported according to 2020 PRISMA statement [17].

### Eligibility criteria

Peer-reviewed studies will be eligible for inclusion if they enrolled community-dwelling adults (≥ 18 years) who were previously admitted to the ICU for ≥24 hours and participated in a post-hospital physical rehabilitation program compared to any control intervention (Table 1). Our primary outcome is physical function, as measured by a validated outcome measure. Physical function refers to an individual's ability to complete physical movements or actions necessary to maintain independence in living [18]. Only randomized controlled trials will be eligible, including mixed methods studies in which the quantitative component was designed as a randomized controlled trial.

### Information sources

We will search Medline (OVID interface, 1946 onwards), Embase (OVID interface, 1974 onwards), Emcare (OVID interface, 1995 onwards), Cumulative Index to Nursing and Allied Health Literature (EBSCOhost interface, 1981 onwards), and Web of Science (Clarivate interface, 1976 onwards) from their respective inceptions. We will hand search reference lists of included studies to identify potentially eligible reports which were missed by our database search.

### Search strategy

We will develop search strategies for each database in consultation with a health sciences librarian. Search strategies will be composed of controlled vocabulary (i.e., subject headings) and keywords of three concepts: (i) critical illness, (ii) physical rehabilitation, and (iii) randomized controlled trial. We will combined search terms using Boolean operators. Each database search strategy will be validated by determining if it located three relevant citations previously identified by the authors [19–21]. Our preliminary Medline search strategy is available in Table 2.

### Screening and extraction

Search results will be uploaded to Covidence systematic review software (2025, Veritas Health Innovation Ltd., Melbourne, Australia) where citations will be automatically deduplicated. Two reviewers, working independently and in duplicate, will complete title and abstract screening and full-text review. During study title and abstract screening, reviewers will assess each title and abstract as potentially eligible or not eligible. If at least one reviewer deems a citation potentially eligible, it will be advanced to study selection. During study selection, reviewers will assess each full report for eligibility.

**Table 1. Eligibility criteria.**

|  | Inclusion | Exclusion |
|---|---|---|
| Population | Community-dwelling adults (≥18 years)<br>Previously admitted to ICU for ≥24 hours<br>Any diagnosis or ICU exposures (e.g., mechanical ventilation, vasopressors or ionotropic agents, hemodialysis) | Pediatric and neonates |
| Intervention | Any post-hospital physical rehabilitation<br>• Must have included a physical component (e.g., strength, aerobic, balance and/or functional training)<br>• May have been initiated before or after acute care discharge<br>• Must have been initiated within 2 years of acute care discharge | Physical rehabilitation conducted solely during acute care admission |
| Comparator | Any comparator (e.g., usual care, education, no formal intervention) | None |
| Outcomes | Primary outcome:<br>• Physical function<br>Secondary outcomes:<br>• Adverse events<br>• Caregiver burden<br>• Cognition<br>• Health care utilization (e.g., emergency department visits, hospital readmissions)<br>• Mental health (i.e., anxiety, depression, post-traumatic stress disorder)<br>• Mortality<br>• Pain<br>• Quality of life<br>• Return to work<br>Timepoints after acute care discharge[a]:<br>• 3-, 6-, 12-month, 2-, 5-, 7-year | N/A |
| Study design | Randomized controlled trial including mixed method studies in which the quantitative component was designed as a randomized controlled trial | Quasi-experimental, cluster and crossover randomized controlled trials |
| Miscellaneous | Peer-reviewed | Abstracts, protocols, editorials, reviews, grey literature |

ICU = intensive care unit; N/A = not applicable.

[a] Timepoints may be refined based on those reported in eligible studies.

Where disagreements exist, they will be resolved through discussion or with a third reviewer to arbitrate. To optimize reviewer consistency, a calibration activity of five to ten publications will be conducted prior to commencement of both study screening and selection. Within the resultant manuscript, study selection will be summarized using a PRISMA flow diagram.

We will complete data extraction independently and in duplicate using a standardized extraction template in Covidence. We will resolve discrepancies through discussion or a third reviewer, as needed. For each included study, we will extract data from its main manuscript and any supplemental material or cited protocols. Where discrepancies exist between documents, we will prioritize information from the main manuscript. Where results are incompletely reported and clarification is required, we will attempt to contact relevant study authors by email up to three times. Extracted information will include:

a. Study characteristics (e.g., title, journal, year of publication, first author, contact information, study design, funding sources)

b. Participant characteristics (e.g., eligibility criteria, sample size, age, sex, ICU admission diagnosis, duration of ICU admission, comorbidities)

c. Intervention characteristics (e.g., setting, intervention description including frequency, intensity, timing, and type, treatment fidelity)

**Table 2. Preliminary Medline search strategy.**

| Concept | Search strategy |
|---|---|
| Critical illness | 1. exp Critical Illness/<br>2. critical* ill*.mp.<br>3. exp Critical Care/<br>4. critical care.mp.<br>5. exp Intensive Care Units/<br>6. (intensive care or burn unit* or coronary care unit* or respiratory care unit* or ICU or ICUs).mp.<br>7. exp Airway Management/<br>8. (airway management or extubat* or intubat* or ventilator* or mechanical* ventilat* or tracheostomy or artificial respiration).mp.<br>9. exp hospital to home transition/ or exp patient discharge/<br>10. patient discharge.mp.<br>11. hospital to home.mp.<br>12. community-dwelling.mp.<br>13. hospital discharge.mp.<br>14. ICU survivor*.mp.<br>15. 1 or 2 or 3 or 4 or 5 or 6 or 7 or 8 or 9 or 10 or 11 or 12 or 13 or 14 |
| Physical rehabilitation | 16. exp Physical Therapy Specialty/<br>17. exp Exercise Therapy/<br>18. exercis*.mp.<br>19. exp Physical Therapists/<br>20. (physical therap* or physiotherap*).mp.<br>21. Rehabilitation/<br>22. rehab*.mp.<br>23. exp Aftercare/<br>24. 16 or 17 or 18 or 19 or 20 or 21 or 22 or 23 |
| Randomized controlled trial | 25. Random Allocation/<br>26. exp randomized controlled trial/<br>27. random*.mp.<br>28. 25 or 26 or 27 |
| Combining concepts | 29. 15 and 24 and 28 |

d. Comparator characteristics (e.g., setting, comparator description including frequency, intensity, timing, and type, treatment fidelity)

e. Outcomes (e.g., construct assessed, outcome measure, assessment timepoint)

f. Sample size, measures of central tendency and dispersion for sought outcomes.

If a study utilizes multiple outcome measures to assess a particular construct, we will prioritize extraction according to their primary outcome measure, followed by the outcome measure with the largest proportion of reported participant data.

## Risk of bias

We will conduct risk of bias assessments for each extracted outcome using version 2 of the Cochrane risk-of-bias tool for randomized trials (RoB 2) [22]. Accordingly, each outcome will be assessed for bias arising from: (i) the randomization process, (ii) deviations from intended interventions, (iii) missing outcome data, (iv) measurement of the outcome and (v) selection of the reported results [22]. Risk of bias will be assessed as 'low risk', 'some concerns,' and 'high risk.' Assessments will be conducted independently and in duplicate with disagreements resolved through discussion or with a third reviewer, as necessary. Within the resultant manuscript, risk of bias assessments will be presented in tables and figures.

## Data analysis and synthesis

**Data synthesis.** Results will be presented in text and table form. If results are sufficiently homogeneous, we will conduct meta-analyses using random-effects modeling with Review Manager (RevMan) software. If results are not homogeneous, we will summarize narratively.

**Measures of treatment effect.** We will report dichotomous variables as risk ratio and 95% CIs. Where authors use the same continuous outcome measure, we will report as mean difference with 95% CIs. Conversely, if similar continuous outcome measures are utilized, we will report as standardized mean difference and 95% CIs.

**Management of missing data.** If data are insufficiently reported to allow for inclusion in a meta-analysis, we will attempt to contact study authors up to three times for clarification. If further clarification is not obtained, we will exclude these results from the meta-analysis and report them narratively.

**Assessment of heterogeneity.** We will assess clinical and statistical heterogeneity. When assessing clinical heterogeneity, we will consider population characteristics, intervention and comparator group components and outcome measures utilized. Statistical heterogeneity will be assessed by visual inspection of forest plots, with the Chi-squared test ($p < 0.1$) and the $I^2$ statistic (0–40% = not important heterogeneity; 30%–60% = moderate heterogeneity; 50–90% = substantial heterogeneity; 75%–100% = considerable heterogeneity) [16].

**Subgroup analyses.** We plan to conduct subgroup analyses according to the following criteria:

a. By physical rehabilitation type (i.e., strength-based versus aerobic-based versus multi-component)

b. By time of physical rehabilitation initiation (i.e., early initiation [within 1 month of acute care discharge] versus delayed initiation [3 months or later after acute care discharge])

c. By primary ICU admission diagnosis (i.e., sepsis versus cardiac versus respiratory versus neurological)

d. By treatment fidelity (i.e., high [≥80% of planned dose] versus low [<80% of planned dose])

**Sensitivity analyses.** We have no plans to conduct sensitivity analyses.

## Quality of evidence

We will assess the strength of the body of evidence for each outcome using the Grading of Recommendations Assessment, Development and Evaluation (GRADE) [23]. Accordingly, results for each outcome will be assessed against five domains: risk of bias, inconsistency, indirectness, imprecision and publication bias [23]. Certainty of evidence will be graded as high, moderate, low or very low and will be presented in a summary of findings table.

## Dissemination of results

We plan to submit our findings for presentation at a critical care related conference and for publication within a peer-reviewed journal. We will also publish our extracted data on Open Science Framework at the time of publication.

## Patient or public involvement

Our aim is to conduct a systematic review that maximizes impact on people who have survived critical illness. As such, the study team includes a patient partner (CC) to guide our approach and conduct. To date, our patient partner has assisted with refining our research questions, eligibility criteria and outcomes of interest [24]. Future roles will include guiding extraction components, interpreting results and identifying future directions from our findings.

## Significance

Approximately 40% of people with critical illness develop ICU-acquired weakness [8]. An international multiprofessional expert panel recently proposed a research agenda related to management of ICU-acquired weakness which included a need to understand how follow-up care models can optimize long-term recovery after critical illness [25]. A recent synthesis indicated the potential post-hospital rehabilitation has to improve physical function in survivors of critical illness [12], however, eligibility was limited to those who required invasive mechanical ventilation, which represents about one-third of ICU patients [14]. Our review will provide a comprehensive synthesis of how post-hospital rehabilitation can influence long-term recovery of survivors of critical illness by including studies which enrolled all patients who were admitted to the ICU for at least 24 hours. Our approach should provide a broad understanding of the impact of post-hospital rehabilitation on survivors of critical illness.

## Supporting information

**S1 File. PRISMA-P Checklist.**
(DOCX)

## Acknowledgments

CF is partially funded by the Canadian Lung Association Allied Health Research Fellowship to complete this work. The Canadian Lung Association had no part in the study design of this protocol and will have no role in data collection and analysis, decision to publish or preparation of the resulting systematic review manuscript.

## Author contributions

**Conceptualization:** Christopher Farley, Anastasia N.L. Newman, Stuart M. Phillips, Jenna Smith-Turchyn, Dina Brooks.

**Funding acquisition:** Christopher Farley, Dina Brooks.

**Methodology:** Christopher Farley, Anastasia N.L. Newman, Christine Caron, Kimia Honarmand, Stuart M. Phillips, Jenna Smith-Turchyn, Dina Brooks.

**Project administration:** Christopher Farley.

**Resources:** Dina Brooks.

**Supervision:** Stuart M. Phillips, Jenna Smith-Turchyn, Dina Brooks.

**Visualization:** Christopher Farley.

**Writing – original draft:** Christopher Farley.

**Writing – review & editing:** Christopher Farley, Anastasia N.L. Newman, Christine Caron, Kimia Honarmand, Stuart M. Phillips, Jenna Smith-Turchyn, Dina Brooks.

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
