## [Decision Letter · Decision Letter 0]

30 Jan 2026

Post-hospital physical rehabilitation for physical function recovery among community-dwelling survivors of critical illness: A systematic review protocol

PONE-D-25-65560

Dear Dr. Brooks,

Your submission has been deemed acceptable for publication. Note that one reviewer suggested changes to your RCT filter in their comments, however my impression is that your filter is quite standard and that no changes are in fact needed. Nevertheless, their comments are appended below for your reference.

**Jiawen Deng**

Temerty Faculty of Medicine, University of Toronto

Please see additional information from the journal below:

Reviewers' comments:

Reviewer's Responses to Questions

**Comments to the Author**

1. Does the manuscript provide a valid rationale for the proposed study, with clearly identified and justified research questions?

Reviewer #1: Yes

Reviewer #2: Yes

2. Is the protocol technically sound and planned in a manner that will lead to a meaningful outcome and allow testing the stated hypotheses?

Reviewer #1: Partly

Reviewer #2: Yes

3. Is the methodology feasible and described in sufficient detail to allow the work to be replicable?

Reviewer #1: No

Reviewer #2: Yes

4. Have the authors described where all data underlying the findings will be made available when the study is complete?

Reviewer #1: Yes

Reviewer #2: Yes

5. Is the manuscript presented in an intelligible fashion and written in standard English?

Reviewer #1: Yes

Reviewer #2: Yes

You may also provide optional suggestions and comments to authors that they might find helpful in planning their study.

Reviewer #1: Dear authors,

Thank you for this submission. I focus my peer review on the search strategies. First, it is reassuring that you have consulted a medical librarian.

I expected at the protocol stage that you provide most if not all search strategies to allow an early peer review on this essential part of your systematic review.

In the current Medline Ovid search strategy, you need to adapt the filter of RCTs. You can consult the filters in the Cochrane handbook or (Glanville 2020).

https://www.cambridge.org/core/journals/international-journal-of-technology-assessment-in-health-care/article/op441-testing-the-sensitivity-and-precision-of-the-cochrane-medline-randomized-controlled-trial-search-filters/A671224D44E3B4B3D9163A1254C58BE3

Reviewer #2: I look forward to this work which will continue to shed light on this expanding area of knowledge and led to further understanding of this field

**Do you want your identity to be public for this peer review?** For information about this choice, including consent withdrawal, please see our Privacy Policy

Reviewer #1: No

Reviewer #2: **Yes:** Douglas Falconer Naylor, Jr., MD, FACS, MCCM

---

## [Editor Report · Acceptance letter]

PONE-D-25-65560

PLOS One

Dear Dr. Brooks,

I'm pleased to inform you that your manuscript has been deemed suitable for publication in PLOS One. Congratulations! Your manuscript is now being handed over to our production team.

Kind regards,

on behalf of

Dr. Jiawen Deng

Academic Editor

PLOS One